# Lung Transplantation in Idiopathic Pulmonary Fibrosis

**DOI:** 10.3390/medsci6030068

**Published:** 2018-08-23

**Authors:** Rosalía Laporta Hernandez, Myriam Aguilar Perez, María Teresa Lázaro Carrasco, Piedad Ussetti Gil

**Affiliations:** Hospital Universitario Puerta de Hierro de Majadahonda Madrid, 28222 Madrid, Spain; myriam.aguilar@salud.madrid.org (M.A.P.); mlazaro.hpth@salud.madrid.org (M.T.L.C.); pied2152@separ.es (P.U.G.)

**Keywords:** lung transplant, idiopathic pulmonary fibrosis, interstitial lung disease

## Abstract

Despite the advances in recent years in the treatment of idiopathic pulmonary fibrosis (IPF), it continues to be a progressive disease with poor prognosis. In selected patients, lung transplantation may be a treatment option, with optimal results in survival and quality of life. Currently, pulmonary fibrosis is the main cause of lung transplantation. However, mortality on the waiting list of these patients is high, since many patients are referred to the transplant units with advanced disease. There is not a parameter that can predict the survival of a specific patient. Different variables are to be considered in order to decide the right time to send them to a transplant unit. It is also very difficult to decide when to include these patients on the waiting list. Every patient diagnosed with IPF, without contraindications for surgery, should be referred early to a transplant unit for assessment. A uni or bilateral transplantation will be decided based on the characteristics of the patient and the experience of each center. The post-transplant survival of recipients with IPF is lower than that observed in other diseases, such as cystic fibrosis or chronic obstructive pulmonary disease as a consequence of their older age and the frequent presence of associated comorbidity. Post-transplant follow-up must be tight in order to assure optimal level of immunosuppressive treatment, detect complications associated with it, and avoid graft rejection. The main cause of long-term mortality is late graft dysfunction as a consequence of chronic rejection. Other complications, such as infections and tumors, must be considered.

## 1. Introduction

The first successful lung transplant in a patient with pulmonary fibrosis as performed by Cooper in 1983. The patient survived with good quality of life for more than five years [1]. Since then, lung transplantation has gradually been introduced for other diseases. Nowadays, all patients with advanced lung disease can be a candidate for lung transplantation, when certain requirements are met [2].

In the past few years there have been considerable advances in the management of patients with IPF. We currently have drugs that can extend survival by slowing disease progression. However, lung transplantation remains the only therapeutic option to restore lung function and improve quality of life and survival.

Historically, chronic obstructive pulmonary disease (COPD) as been the main indication for lung transplantation. In recent years the preponderance of COPD as an indication for transplantation has been replaced by pulmonary fibrosis [3]. This change has occurred as a result of the development and application of new criteria for the waiting list prioritization. The Lung Allocation Score (LAS) considers both the risk of dying while waiting and the survival increase after transplantation. The Lung Allocation Score is calculated from clinical and physiological variables such as age, underlying disease, % forced vital capacity (FVC%), six-minute walk test (6MWT), and pulmonary artery pressure. Patients are prioritized on the waiting list according to LAS. The LAS is applied in the USA, Germany and the other countries that constitute the Eurotransplant International Foundation (Eurotransplant). Under the LAS system, there are some data from the US, mainly in pulmonary fibrosis patients, that describe a decrease in wait list time [4]. Since the implementation of LAS, the percentage of patients included on the waiting list with pulmonary fibrosis has increased in the US from 33.8% to 46%, and the transplantation procedures from 21% to 32% [5].

In Spain, pulmonary fibrosis is also the main indication for lung transplantation. According to data from the Spanish National Transplant Organization (ONT), in 2017, 36% of the patients included on the waiting list and 39% of the recipients, had pulmonary fibrosis [6].

Regardless of the criteria used for patient prioritization, the transplantation teams take a great responsibility when offering lung transplant. They have to follow ethical criteria of utility, justice, and benefit. To fulfill this, it is essential to choose patients that really need transplantation and would expect high benefit from it.

In this chapter, the characteristics of pulmonary fibrosis candidates will be reviewed, in addition to their selection criteria and pretransplant management. The most frequent post-transplant complications and results after surgery will also be discussed.

## 2. Pretransplant Evaluation

Pulmonary transplantation should be considered for patients with pulmonary fibrosis who meet the selection criteria and do not have significant contraindications for its performance (Table 1). In order to optimize transplant results, it is important to identify the adequate time of referral to a lung transplant unit and accomplish the best management before the intervention.

### 2.1. Selection of Patients

Patients with pulmonary fibrosis eligible for transplantation are those who have a high risk of death (>50%) in the following two years without transplantation, high probability of surviving at least 90 days after the intervention (>80%), and high probability of surviving five years after the transplant (>80%) from any other medical condition [2].

It is important to individually assess survival expectancy with and without transplant, being aware of significant comorbidity that may worsen the expected result. The main absolute contraindications for transplantation are recent neoplasia, dysfunction of another vital organ, uncorrected coronary disease, clinical instability, and extreme obesity. 

A prior history of poor treatment compliance is an important consideration. Lung transplantation is a very complex and demanding process, for both the patient and family. It is essential to adequately inform the patient about the procedure and its results, as well as ensure their active participation in the process and adherence to treatment. 

Lung transplantation is a complex and high-risk procedure. All the relative contraindications have to be compiled in order to consider the risk of peri-operative complications (Table 1). Pulmonary fibrosis usually affects elderly people, with an average age of 66 years at the time of diagnosis [7]. Given that age over 65 is a relative contraindication for transplantation, a large proportion of patients with pulmonary fibrosis are diagnosed close to the borderline age. This older age compromises the transplant results, especially in the presence of other associated comorbidities, as cardiovascular problems, that may negatively affect the short and long-term prognosis [8].

The frequency of coronary heart disease in patients with idiopathic pulmonary fibrosis (IPF) varies between 4% and 25%. Some authors have postulated that the pro-inflammatory state of IPF is associated with a higher frequency of cardiovascular disease [9]. It is recommended to perform coronary angiography in the evaluation of candidates aged >55 years with associated cardiovascular risk factor such as smoking, dyslipidemia, or diabetes. Pretransplant coronary angiography can detect coronary lesions in more than half of these candidates [10]. In our experience, coronary angiography detected lesions in the 25% of cases. It was also observed that coronary disease was more frequent in patients with IPF than in patients with COPD [11]. 

The frequency of pulmonary hypertension (PH) in patients with IPF is not well known and depends on the method used for diagnosis. Patients with PH more frequently require external cardiorespiratory support during surgery and develop more perioperative complications [12].

Gastroesophageal reflux is frequent in patients with IPF with an estimated prevalence of 50–85%. Gastroesophageal reflux surgery before transplantation can be a preventive measure against chronic rejection. It should be considered when symptoms cannot be controlled with medical treatment [13,14].

Osteoporosis is very frequent in patients with advanced lung disease. Although it has no direct impact on mortality, it may increase the risk of fractures and impair quality of life after transplantation.

The nutritional status has prognostic value in lung transplant patients. Malnutrition, defined by body mass index (BMI) below 17, increases the mortality while on the waiting list. Malnutrition is also associated with more need for mechanical ventilation and longer stay in the intensive care unit after transplantation [15]. Overweight (BMI > 25) increases cardiovascular risk and perioperative complications. BMI > 30 is an independent risk factor for early mortality [16,17].

Recently, short telomere length has also been shown to be associated with a shorter life expectancy in IPF [18]. Approximately 25% of sporadic IPF and 37% of family related cases have telomere shortening with mutations on telomerase RNA component *TERC* and telomerase reverse transcriptase *TERT* [19]. These mutations are also associated with extra-pulmonary pathology that may compromise evolution after transplantation. It has been observed that transplant recipients with telomere shortening present worse survival and less free time of chronic graft dysfunction [20,21]. Some centers search for telomere length shortening in young patients and family related cases of IPF [22,23]. However, the clinical implications of the telomere length mutations in the post-transplant evolution are still to be defined. Therefore, routine telomere analysis is not considered necessary in the pretransplant evaluation.

### 2.2. Transplantation Window

It is important to differentiate two clinical decisions in the evolution of the IPF: when to refer the patient to a transplant unit, and when to include him/her on the transplant waiting list. The criteria for referral have been defined taking into account the potential gain in survival after transplantation [2]. 

Five-year post-transplant survival is close to 50% [8]. The mean life expectancy of IPF patients after diagnosis is 3 years and five-year survival is around 30–35% [24]. Taking into account these data, it is evident the survival gain that transplantation brings to these patients. Pulmonary fibrosis patients should be referred early, when forced vital capacity (FVC) < 80% and/or diffusing capacity of the lung for carbon monoxide (DLCO) < 40% (Table 2).

It is more difficult to decide when to include a patient on the waiting list. The course of pulmonary fibrosis is very variable; some patients have prolonged survival, while others deteriorate rapidly. All prognostic factors should be taken into account, in addition to the potential deterioration while waiting for transplantation and the expected delay. Despite the fact that the application of LAS has reduced the waiting time and increased the number of procedures for IPF, the mortality of these patients on the waiting list is still higher (14–67%) than that observed in other diagnostic groups [5].

Studies that assess prognostic factors for pulmonary fibrosis have focused on two fundamental aspects: the baseline characteristics at the time of diagnosis and the disease progression. In relation to the baseline characteristics, the criteria of worse prognosis are: gender male, older age, hospitalization due to respiratory causes, low body mass index, and extensive radiological involvement. Functional data of worse prognosis are: FVC < 80%, DLCO < 40%, distance walked < 250 m and/or oxygen saturation values < 88% during the 6MWT, and pulmonary hypertension [25,26].

The evolution of lung function during follow-up has a higher prognostic value than the baseline values. Six months mortality predictors are: decrease of the FVC > 10%, DLCO > 15%, or decrease over 50 m in the 6MWT [25,26]. Other factors of poor prognosis during follow-up are pulmonary hypertension and hospitalization due to functional deterioration, pneumothorax, or exacerbation of the disease (Table 2). 

No single variable is able to accurately predict the survival of IPF patients, so multidimensional models of prediction have been developed. The Composite Physiological Index (CPI) takes into account physiological variables and the existence of emphysema. Predictive value for mortality with CPI was higher compared to individual functional variables. Subsequently, other scales have been developed combining variables from different domains, to better identify the severity of pulmonary fibrosis. These include GAP (gender, age, physiology), which assess sex, age, and physiological variables such as FVC% and DLCO% and the Risk Stratification Score (RISE) based on the Medical Research Council Dyspnea Score (MRCDS), 6MWT, and CPI. Both GAP and RISE scales have shown to be better predictors of mortality than the parameters assessed individually. These three multidimensional scales, CPI, GAP, and RISE are useful for mortality prediction of patients with IPF, but their role in patients undergoing lung transplantation has been poorly evaluated.

Fisher et al. [27] analyzed the role of GAP, RISE, CPI, and LAS in the prediction of mortality in IPF patients assessed for lung transplantation. They observed that the multidimensional specific IPF scales had similar sensitivity and specificity for mortality prediction such as LAS. 

The multidimensional specific IPF scales can be useful for deciding when to refer a patient to a transplantation unit and when to include him/her on the waiting list. The GAP-calculator allows classification of patients into three mortality risk groups (Table 3). At stage I, GAP score 0–3, the estimated risk of mortality in the first year is low; 5.2%, lower than the mortality associated with transplantation. At stage II, GAP score 4–5, the estimated risk of mortality is 16.9%, similar to transplantation. In this stage, the decision to include on the waiting list must be individualized, taking into account the preferences of the patient and the waiting time estimation. At stage III, GAP score 6–8, the estimated risk of mortality is high; 41.7%, so there should be no doubts regarding the benefit of transplantation. 

For example, for a 64 year-old male patient, with FVC 70%, DLCO 40%, GAP index 4, stage II, we would recommend referral and close follow-up. For the same patient, with FVC < 50% and DLCO < 35%, GAP index 6, stage III, prompt inclusion on the transplantation list would be recommended. The predictive capacity of GAP has been improved including follow-up variables such as FVC evolution and hospitalization due to exacerbation [28].

### 2.3. Preoperative Management

The international consensus for the management and treatment of IPF establishes grade A therapeutic recommendation to use of antifibrotic drugs, pirfenidone and nintedanib, with level 1 of scientific evidence, in patients with mild or moderate functional impairment [29].

The efficacy and safety of antifibrotic drugs in patients with advanced disease waiting for lung transplant is poor and is based on the analysis of clinical cases. Due to their antifibrotic action, both drugs could affect the surgical wound healing or cause anastomotic complications [30]. In addition, nintedanib, as an inhibitor of tyrosine kinases platelet-derived growth factor and vascular endothelial growth factor receptors, may increase the risk of perioperative bleeding. The European Medicine Agency recommends that nintedanib should be discontinued before a major surgery [31]. Patients that are waiting for transplant usually have only hours notice of the upcoming surgery. In these patients, the treatment discontinuation only is possible right before transplantation. Several transplantation units recommend holding antifibrotic treatment until the patient is called for transplantation. Antifibrotic drugs do not seem to increase post-transplant complications, such as dehiscence of the bronchial anastomosis, perioperative bleeding, or graft dysfunction [32,33]. On the other hand, the discontinuation of antifibrotic treatment in patients on the waiting list can worsen the clinical situation of the patient. When slowing the progression of the disease with antifibrotic treatment, the patient is more likely to survive on the waiting list and get to transplantation in better condition. The general recommendation is not to interrupt the antifibrotic treatment before transplantation [34,35].

There is no clinical evidence to justify treatment with corticosteroids in IPF. Their use is currently limited to two situations: acute exacerbation and treatment of some difficult symptoms, such as incoercible coughing. It would be prudent to reduce dose or discontinue corticosteroid treatment when IPF patients are assessed for lung transplantation [36,37,38].

Patients with myopathy due to lack of physical exercise, terminal respiratory failure, or use of corticosteroids have a greater need for more prolonged mechanical ventilation after transplant and have a worse immediate post-transplantation prognosis. Immobilization or sedentary lifestyle is also considered a risk factor for cardiovascular events. Therefore, an adequate assessment by a rehabilitation team is necessary to improve the muscular function before transplantation.

## 3. Type of Transplant

Unilateral transplantation has historically been considered the elective type of transplant in patients with pulmonary fibrosis. Recently, there is a clear tendency to perform bilateral transplant in these patients. While in 1991 only 15% of patients with pulmonary fibrosis received a bilateral transplant, in 2016 this percentage exceeded 50%. The reasons for this change are not clear. The International Society for Heart and Lung Transplant (ISHLT) registry data provides better survival for bilateral transplantation; 7 years versus 4.5 years [8]. However, it is not clear if this improvement of survival is related to the type of transplant as it can also be explained by the different characteristics of the recipients selected for unilateral or bilateral lung procedure.

There is poor quality evidence from studies that compare the results of unilateral versus bilateral transplant, to recommend the use of either procedure. Therefore, in absence of a guidance based on scientific evidence, the choice of transplant type will depend on the preferences of the different transplantation groups [39]. These preferences are usually based on the experience of each group, the age of the recipient, the difference in perfusion between both lungs, the existence of significant comorbidity. or pulmonary hypertension.

A recent article [40] has reviewed the different aspects to take into account before choosing the type of procedure. The optimization of the organs availability is an important consideration, since the unilateral lung transplantation can benefit two recipients. The number of patients on the waiting list far exceeds the number of organs available. Pulmonary fibrosis patient mortality while waiting for transplantation is higher than in other diseases, despite its reduction after the introduction of LAS. An additional advantage of the unilateral transplant is the shorter time on the waiting list, which can reduce the risk of mortality [41]. 

In terms of outcome in lung function and survival, both procedures have advantages and disadvantages. The unilateral transplant presents fewer perioperative complications than the bilateral one, due to the greater simplicity of the procedure. With bilateral transplantation long-term survival is greater due to greater functional reserve and avoidance of possible complications in the native lung, such as infection and neoplasia [42,43,44]. 

General recommendations cannot be established since most of the data is based on retrospective series, so transplant type decision should be individualized. Unilateral pulmonary transplantation is a feasible choice in patients with pulmonary fibrosis, mostly if they associate comorbidity and high risk of mortality on the waiting list.

## 4. Results

Lung transplant recipients experience improvements in lung function, quality of life, and survival after the intervention.

### 4.1. Functional Evolution

The maximum pulmonary function reached after a bilateral transplantation is independent of the underlying disease. It depends on the characteristics of the graft, the recipient thoracic cage, and postoperative complications. Three months after surgery FVC and forced expiratory volume in the first second, values in bilateral recipients can be higher than 80% of predicted values and reach 100% after 6–12 months [45,46].

In unilateral transplant recipients, lung function stabilizes earlier, around the third month, because there is less surgical trauma [47]. The maximum pulmonary function reached is usually lower than in recipients of bilateral transplantation and depends on the characteristics of residual lung. In recipients affected by interstitial diseases the residual lung tends to collapse. After three months FVC and FEV1 can get over 80% of predicted values (Figure 1).

### 4.2. Survival

The survival after lung transplantation is worse than other solid organ transplants such as cardiac, renal, or hepatic. This is because on the long-term survival, there is more frequent development of chronic rejection. Advances in perioperative management and immunosuppression have improved early results but have less impact on long-term survival. According to the data of the ISHLT registry, the mean post-transplant survival is six years, with a global survival of 80% in the first year, 64% in the third year, 54% in the fifth, and 32% at the tenth year [8]. The mean survival for patients who survive the first year is 8.1 years. These data are similar to those obtained in the ONT Spanish registry, with non adjusted survival at three months, first year, and third year of 79%, 71%, and 60%, respectively [48].

Lung transplantation improves survival of patients with pulmonary fibrosis [49]. The average post-transplant survival of these patients is lower than other diagnostic groups such as cystic fibrosis or COPD. Pulmonary fibrosis patients have higher mortality within 90 days and in the first year after transplantation [50,51,52].

Early graft dysfunction and non-cytomegalovirus infections are the main causes of death during the first year after transplantation. Chronic lung allograft dysfunction (CLAD) is responsible for 30% of annual deaths after the first year. The development of malignant neoplasia becomes relevant after the first year of the transplantation, hitting approximately 10% of deaths.

### 4.3. Quality of Life

Different studies have observed that quality of life for the transplant recipients is better than candidates. This improvement is usually maintained until the development of chronic rejection [53].

Transplanted patients have better physical and psychological situation than the candidates on the waiting list. Nevertheless, both present more disturbance of self-esteem, anxiety, and depression, compared to the normal population.

Singer et al. [54] assessed the impact of LAS introduction in quality of life results, since older and more severe patients underwent transplantation. They observed that transplantation continued to offer an improvement in quality of life. Also, they recommended that limiting factors should be identified in certain subgroups, such as recipients over 65 years, in order to maximize the net benefits of the transplant.

## 5. Post-Transplant Monitoring

The goals of post-transplant follow-up are to maintain the optimal degree of immunosuppression, deal with the multiple adverse effects of immunosuppressive drugs, and diagnose and treat post-transplant complications as soon as possible. 

### 5.1. Follow Up

Spirometry is the main tool for follow up. It is performed periodically and routinely in these patients. Pulmonary function improves after transplantation, so baseline FEV1, calculated as the mean of the two best consecutive values, has to be established and must be recalculated periodically. The deterioration in FEV1 can indicate multiple problems such as acute or chronic rejection, infection or hyperinflation of the native lung [55]. 

Home monitoring of lung function is very useful for early detection of graft worsening [56]. When there is a decrease of more than 10% of the baseline values of FEV1, the patient should alert his reference center. If the decline in lung function is confirmed with conventional spirometry, a diagnosis evaluation to monitor graft status should be performed; including chest radiology and fibrobronchoscopy with bronchoalveolar lavage and transbronchial biopsy [57,58]. Complications of the bronchial suture, episodes of rejection, and certain infections or colonization, require early treatment to minimize graft damage.

### 5.2. Complications

After the immediate postoperative period, complications come from immunosuppression and drug toxicity (Table 4). Among the most frequent complications during the first year after transplantation are infections and acute rejection. After the first year the main complications are neoplasia and CLAD [59].

#### 5.2.1. Infections

The infections are one of the most important causes of both early and late mortality. Infections are due to the permanent contact of the graft with the environment, through the inspired air. The high immunosuppression level necessary to avoid acute and chronic rejection and the abolition of the cough reflex, as a consequence of the bronchial denervation, favors the persistence of inhaled microbes. In IPF recipients, infections account for 56% of all complications, with a 15% mortality in the first six months after surgery. 

Regarding the chronology of infections, the most frequent in the immediate post-transplantation period and during the first two months, are bacterial infections. The most frequent organism is *Pseudomonas aeruginosa*, followed by *Staphylococcus aureus*. Viruses acquire prominence, after the first month, especially cytomegalovirus (CMV*).* Fungal infections can appear in the immediate postoperative period and after the first month, along with *Pneumocystis jiroveci* and tuberculosis.

The high frequency of bacterial infections requires the establishment of preventive treatment with antimicrobial prophylaxis. The antibiotic selection is based on the microorganisms isolated in the recipient and in the donor’s bronchial aspirate, obtained before or during transplantation.

Cytomegalovirus is the second cause of infection in patients with lung transplantation. Cytomegalovirus can be acquired by transmission from the donor, by blood transfusion, or by reactivation of a latent infection in the recipient. The greatest risk of CMV infection is seen when the donor is seropositive and the recipient is seronegative for CMV*.* Cytomegalovirus infection causes different clinical syndromes: flu-like syndrome, pneumonia, blood cytopenia (leukopenia and thrombocytopenia), digestive disorders, and hepatitis. In recipients at risk, prophylaxis with ganciclovir in the first three months after transplantation significantly reduces the frequency and severity of CMV disease [60].

Fungal infections are a frequent cause of morbidity and mortality in the lung transplant recipients. Lung transplantation is the solid organ transplant with greater risk for fungal infections. The genus *Aspergillus* is responsible for most fungal diseases in these patients, particularly *Aspergillus fumigatus*. The spores of *Aspergillus* are ubiquitous in and out of the hospital. The most common *Aspergillus* infection in the lung transplant recipients is tracheobronchitis [61]. 

*Pneumocystis jiroveci* infection has almost been eliminated by routine prophylaxis with trimethoprim-sulfamethoxazole. This prophylaxis also has a preventive effect against other germs, such as *Nocardia*.

In lung transplant recipients, tuberculosis can develop from latent foci of both donor and recipient. It is advisable to start pretransplant chemoprophylaxis with isoniazid in patients at risk of tuberculosis, determined by Mantoux and/or Quantiferon.

The most frequent side effects of immunosuppressive treatment are arterial hypertension, dyslipidemia, and diabetes mellitus (Table 4). The old age of patients with IPF, which are mostly over 60, and the longevity after transplantation, favor the cardiovascular side effects of the drugs.

#### 5.2.2. Chronic Lung Allograft Dysfunction

The chronic lung allograft dysfunction or chronic rejection is the most important problem that hinders the long-term outcome of lung transplant patients. The probability of developing chronic rejection increases with the post-transplant survival time; affects 12% of the recipients in the first year and more than 50% after five years. The main manifestation of chronic pulmonary rejection is bronchiolitis obliterans syndrome (BOS), which is the most important cause of lung transplant mortality after the first year and is responsible for 30% of annual deaths. For many years, BOS was considered the only manifestation of CLAD. The terms “chronic rejection” and “BOS” were considered synonymous. However, in 2011 a different entity was described restrictive allograft syndrome (RAS). Since then, the term CLAD has been used to refer to all variants of chronic pulmonary dysfunction, mainly BOS and RAS. Restrictive allograft syndrome is defined as an irreversible fall in total lung capacity (TLC) > 10% compared to the baseline values, with fibrotic lesions that occur at the periphery of the lung (visceral pleura, alveolar, and interlobular septa). Bronchiolitis obliterans syndrome is characterized by an obstructive pattern secondary to fibrotic lesions that mainly occur at the bronchiolar level [62,63,64].

Both types of CLAD share some of the risk factors and can coexist. This suggests that the two entities have the same pathogenic pathway. The epithelial cell damage, secondary to immunological and/or non-immunological mechanisms, triggers an initial phase of lymphocytic inflammation of the submucosa. This initial damage is followed by a fibro proliferative phase with degradation of the extracellular matrix, proliferation of fibroblasts, myofibroblasts, antioxidant depletion, aberrant deposit of collagen fibers, and loss of the homeostasis in the regeneration of the extracellular matrix. It has been postulated that the fibrotic lesions of the airway are the result of the transition of epithelial cells to mesenchymal cells (EMT), a mechanism by which, in advanced stages, endoluminal obstruction occurs (Figure 2) [65,66].

Currently, there is no effective therapy for CLAD, so the only possible approach is primary prevention and early detection, before fibrosis has been established. A better definition of risk factors and markers involved in the pathogenesis could help us prevent CLAD development and improve the results of long-term transplantation. In this sense, our transplant group conducted a study, the results of which support the hypothesis that the analysis of bronchoalveolar lavage (BAL) cells, including neutrophils, and cytotoxic lymphocytes (NK and TCD8 cells), together with cytokines interleukin (IL)-8, IL-6, IP-10 (interferon-inducible protein 10), RANTES (regulated on activation normal T-cell expressed and secreted), MIG (monokine induced by gamma) and MCP-1 (monocyte chemoattractant protein 1), as a reflection of profibrotic inflammatory activity, can facilitate the early recognition of CLAD [67].

The different strategies in the treatment of chronic rejection are confusing, since the published studies have small size, heterogeneous sample size, and/or lack a control group. They consist on increasing immunosuppression, minimize the risk of infection with nebulized antibiotherapy, and early treatment of bronchial colonization, especially Pseudomonas. Other therapeutic measures to consider are treatment of gastroesophageal reflux, pulmonary rehabilitation, thymoglobulin, and extracorporeal photoapheresis [68]. In selected young patients with progressive evolution, retransplantation can be considered as an exceptional measure.

Among the possible therapies that are currently under investigation are antifibrotic drugs such as pirferidone and nintedanib. Several cases of favorable response with both antifibrotic drugs have been described in patients with progressive fall in lung function, despite the usual treatments [69,70].

#### 5.2.3. Neoplasia

The risk of developing de novo neoplasia in patients with solid organ transplants is higher than in the nontransplanted population. The incidence is between 4% and 18%. The main risk factor is immunosuppressive treatment, but there are other factors such as the age of the recipient and the post-transplant time, with a frequency of 13% at 5 years and 28% at 10 years [8].

The epidemiology and risk factors for de novo tumor were analyzed retrospectively in two Spanish lung transplant units [71]. Out of a total of 1353 recipients, 125 developed some type of neoplasm after an average of 3.7 years post-transplant. This frequency was five times higher than in the general population. The risk of developing cancer was higher in patients over 55 years, men, and those with a history of smoking over 20 pack-years. The most frequent tumor was skin cancer (32%), followed by lymphoproliferative diseases (18%), and lung cancer (16.5%).

According to data from the ISHLT registry the most important risk factors of de novo skin neoplasia are sun exposure, papilloma virus, and duration of immunosuppression. Lymphomas are mainly associated with new infection of the Epstein-Barr virus (EBV).

Bronchial carcinoma involves 2% of the neoplasias in recipients of lung transplant. The risk factors for the development of primary bronchogenic carcinoma in the native lung are older age, previous heavy smoking history and unilateral transplant (Figure 1). The most common underlying disease is pulmonary fibrosis [72]. In our program, lung neoplasia affected 3.63% of the total transplant population [73]. In almost half of the cases, the underlying disease was pulmonary fibrosis. The mean time between transplantation and cancer diagnosis was 39 months. The majority of the patients had a previous history of smoking and in more than half the tumor developed in the native lung. The most common histological type was adenocarcinoma (56.5%), followed by squamous carcinoma (30.4%). Of the lung tumors, 56.4%, were in advanced stage with metastasis at the time of diagnosis. Survival after first year of diagnosis was 45.64% and global mortality for lung cancer was 70% [73].

## 6. Conclusions

Lung transplant offers survival benefit in carefully selected patients with IPF. The number of lung transplants due to IPF has increased in recent years, mainly after the development of LAS, but waiting list mortality of these patients remains high. Therefore, it is important that the patients diagnosed with IPF be remitted early to transplant units for evaluation. 

The complexity of surgery and post-transplant treatment require a comprehensive assessment of the candidates. There are comorbidities associated with the underlying disease that can negatively affect post-transplant progress, such as cardiovascular diseases and gastroesophageal reflux. They must be detected and corrected prior to surgery to minimize risks. 

The use of antifibrotic drugs in patients on the waiting list may be useful to stabilize the clinical situation. Although the clinical cases with antifibrotic treatment evaluated are scarce, these drugs do not seem to increase post-transplant complications. 

The choice of uni or bilateral transplant depends on the characteristics of the recipient and the preferences of the transplant unit, taking into account the waiting list time and mortality. 

The chronic lung allograft dysfunction represents the main cause of death during long-term follow-up. Post-transplant investigations are necessary to understand the mechanisms involved in the development of CLAD, and how to treat it, as it remains a progressive and incurable complication. Other competing causes of death are infections and the development of de novo neoplasia.

## Figures and Tables

**Figure 1 medsci-06-00068-f001:**
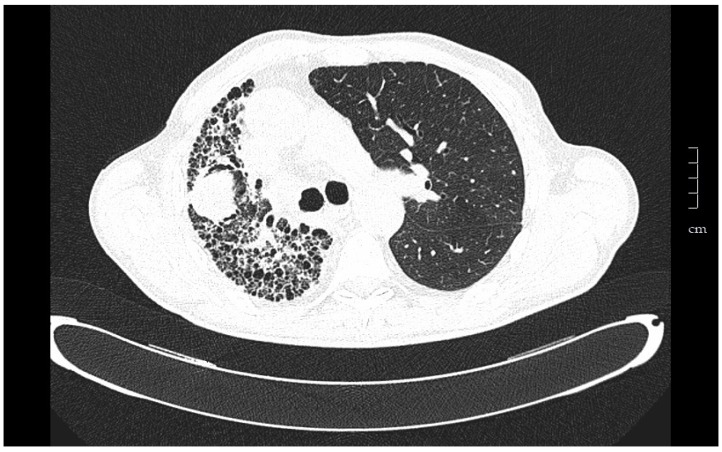
Tomography of a 67-year-old man that underwent left unilateral lung transplantation for pulmonary fibrosis. The native lung tends to collapse, and the graft expands freely. Eight years after transplantation a tumor in the native lung was found in a follow-up chest scan. Ultrasound-guided fine needle aspiration puncture diagnosis was squamous cell carcinoma.

**Figure 2 medsci-06-00068-f002:**
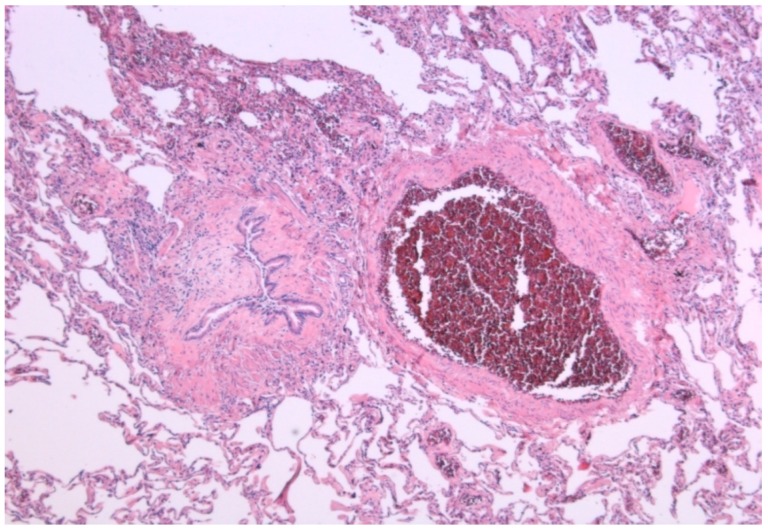
Histological image, magnification 100×, with obliteration of the endobronchial lumen, compared with the vascular lumen, in a patient with chronic rejection (courtesy of Dr. Clara Salas).

**Table 1 medsci-06-00068-t001:** Indications and contraindications of lung transplant.

**Indications**
High risk of death (>50%) in the following two years without transplant
High probability (>80%) of surviving at least 90 days after transplantation
High probability (>80%) of surviving 5 years after transplantation of any general medical condition
**Absolute Contraindications**
Recent tumor history (>5 years free of disease)
Dysfunction of another major organ
Nonvascularizable coronary disease
Hemorrhagic diathesis
Deformities of the chest wall
Morbid obesity
Tuberculosis infection
Infection by highly resistant germs without control
Severely altered functional status with inability to rehabilitate
Severe psychiatric disorders
Poor social support
Nonadherence to treatment
**Relative Contraindications**
Age > 65 years
Obesity (BMI 30–34.9)
Severe malnutrition
Symptomatic severe osteoporosis
Previous thoracic surgery with pulmonary resection
Mechanical ventilation and/or extracorporeal support
Infection by virus B or C with evidence of significant liver injury and/or portal hypertension

BMI: body mass index.

**Table 2 medsci-06-00068-t002:** Criteria for referral and inclusion in the waiting list.

**Criteria for Referral to Transplant Units**
Histopathologic diagnosis of UIP or fibrotic NSIP
FVC < 80% and/or DLCO < 40%
Dyspnea or functional limitation attributable to lung disease
Requirement of oxygen therapy in exercise or at rest
Lack of response to treatment
**Criteria for Inclusion in the Waiting List**
Decrease ≥ 10% of FVC during 6 months of follow-up
Decrease ≥ 15% of the DLCO during 6 months of follow-up
Desaturation < 88% or distance < 250 m during 6MWT or descent > 50 m in the distance walked, during 6 months of follow-up
Pulmonary hypertension
Hospitalization due to deterioration, pneumothorax, or acute exacerbation

UIP: usual interstitial pneumonia; NSIP: nonspecific interstitial pneumonia; FVC: forced vital capacity; DLCO: carbon monoxide diffusing capacity; 6MWT: the six-minute walk test.

**Table 3 medsci-06-00068-t003:** Gender, age, physiology (GAP) index for the transplant consideration.

GAP Score	Stage	One Year Mortality	Recommendation
0–3	I	5.6%	Follow-up
4–5	II	16.2%	Individualize according to evolution and waiting time
6–8	III	39.2%	Include in the waiting list

**Table 4 medsci-06-00068-t004:** Main side effects of immunosuppressive drugs.

Immunosuppressive Drugs	Side Effects
Corticosteroids	Hypertension, hyperlipidemia, hypocalcemia, osteoporosis, weight gain, cataracts, and psychological symptoms
Antimetabolites (mycophenolate, azathioprine)	Myelosuppression, hepatotoxicity, alopecia, gastrointestinal intolerance, acute pancreatitis, infections, and neoplasia
Calcineurininhibitors (cyclosporine, tacrolimus)	Nephrotoxicity, hypertension, hyperuricemia, hyperkalemia, diabetes, hyperlipidemia, neurotoxicity, gingival hyperplasia, infections, and lymphoproliferative disease
Inhibitors of mammalian Target of Rapamycin (everolimus, rapamycin)	Hyperlipidemia, diabetes, edema, proteinuria, and pneumonitis

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
