# Peer review of "Lung Transplantation in Idiopathic Pulmonary Fibrosis"

_medsci, 2018, doi:10.3390/medsci6030068_

Round 1

Reviewer 1 Report

The authors provide a review of lung transplantation in IPF, discussing important considerations prior to referral, indications and absolute/relative contraindications for surgery and the decision making surrounding when best to perform and what type of surgery. Post surgery complications are also addressed.

Whilst the topic is an important one, especially for the general respiratory physician, I feel that the quality of English used precludes its acceptance in its current form. Furthermore, there are numerous typos. 

The manuscript would benefit from an introductory statement that guides the reader as to the aim of the review with  summary or discussion of main conclusions, rather than ending abruptly.

I would also recommend that the paper is written in the third person - removing he/him as descriptors. 

Author Response

-We have rewritten some sentences trying to improve the quality of English. We have searched for typos.

  *In this chapter there will be reviewed the characteristics of IPF candidates for lung transplant, the selection criteria and their pre-transplant management. Also will be discussed the most frequent post-transplant complications and the results after surgery*

-We have added to the manuscript an introductory statement and final conclusions.  

 * Lung transplant offers survival benefit in carefully selected patients with IPF. The number of lung transplants due to IPF has increased in recent years, mainly after the development of LAS, but waiting list mortality of these patients remains high. Therefore, it is important that the patients diagnosed with IPF be remitted early to transplant units for evaluation.   The complexity of surgery and post-transplant treatment require a comprehensive assessment of the candidates. There are comorbidities associated with the underlying disease that can negatively affect post-transplant evolution, such as cardiovascular diseases and gastro-esophageal reflux. They must be detected and corrected prior to surgery to minimize risks.   The use of antifibrotic drugs in patients on the waiting list may be useful to stabilize the clinical situation. Although the clinical cases with antifibrotic treatment evaluated are scarce, this drugs doesn’t seem to increase post-transplant complications.   The choice of the type of transplant, uni or bilateral, depends on the characteristics of the recipient and the preferences of the transplant unit taking into account the waiting list time and mortality.   CLAD represents the main cause of death during long-term follow-up. Post-transplant investigations are necessary to understand the mechanisms involved in the development of CLAD, and how to treat it, as it remains a progressive and incurable complication. Other competing causes of death are infections and the development of de novo neoplasia.*

Reviewer 2 Report

A well written and comprehensive review.

Overall, it was well written and easy to follow. A minor comment would be that the last section (5.2 complications) could use a few more subheading. This is quite a long section and covers infections, drug toxicity, chronic rejection and cancer.

The review also lacks a brief conclusion and abruptly ends. It would be nice to have one for completeness. 

Author Response

- Section 5.2 has been divided into different sections according to the complications described

(5.2.1 Infections, 5.2.2 chronic graft dysfunction, 5.2.3 Neoplasia)

 -At the end you will find the final conclusions.

 * Lung transplant offers survival benefit in carefully selected patients with IPF. The number of lung transplants due to IPF has increased in recent years, mainly after the development of LAS, but waiting list mortality of these patients remains high. Therefore, it is important that the patients diagnosed with IPF be remitted early to transplant units for evaluation.   The complexity of surgery and post-transplant treatment require a comprehensive assessment of the candidates. There are comorbidities associated with the underlying disease that can negatively affect post-transplant evolution, such as cardiovascular diseases and gastro-esophageal reflux. They must be detected and corrected prior to surgery to minimize risks.   The use of antifibrotic drugs in patients on the waiting list may be useful to stabilize the clinical situation. Although the clinical cases with antifibrotic treatment evaluated are scarce, this drugs doesn’t seem to increase post-transplant complications.   The choice of the type of transplant, uni or bilateral, depends on the characteristics of the recipient and the preferences of the transplant unit taking into account the waiting list time and mortality.   CLAD represents the main cause of death during long-term follow-up. Post-transplant investigations are necessary to understand the mechanisms involved in the development of CLAD, and how to treat it, as it remains a progressive and incurable complication. Other competing causes of death are infections and the development of de novo neoplasia.*

Reviewer 3 Report

General comments

Laporta Hernandez R and colleagues described various phases of lung transplantation in idiopathic pulmonary fibrosis (IPF), including pre-transplant evaluation, type of transplant, post-transplant monitoring. The manuscript deals with a relevant topic in respiratory medicine: description of the only treatment in IPF able to improve survival. However, some issues need to be better addressed:

Major comments

The novelty of the findings included in the manuscript is extremely limited: various “review articles” are already available in the literature describing pre-transplant evaluation, type of transplant, post-transplant monitoring (1-3).

The Authors should describe the potential implication of anti-fibrotic treatment, both with nintedanib or pirfenidone, in IPF patients before and after lung transplantation

Minor comments

The manuscript is well-written and accurately edited

References

Kumar A, Kapnadak SG, Girgis RE, Raghu G. Lung transplantation in idiopathic pulmonary fibrosis. Expert Rev Respir Med. 2018 May;12(5):375-385

Brown AW, Kaya H, Nathan SD. Lung transplantation in IIP: A review. Respirology. 2016 Oct;21(7):1173-84.

Kistler KD, Nalysnyk L, Rotella P, Esser D. Lung transplantation in idiopathic pulmonary fibrosis: a systematic review of the literature. BMC Pulm Med. 2014 Aug 16; 14: 139

Author Response

In our manuscript we compile the last evidence published, including 2018. We also have shared our own experience about comorbidities (section 2.1) and complications (5.2.2 CLAD). My 2017 PhD explores new research in the early diagnosis of CLAD.

-Aryal, Nathan. Single vs. Bilateral lung transplantation: when and why. CurrOpin Organ Transplant 2018 Apr 7 (Epub ahead of print)

-Lung cancer in lung transplantation: incidence and outcome.Pérez-Callejo D, Torrente M, Parejo C, Laporta R, Ussetti P, Provencio M.PostgradMed J. 2018 Jan;94(1107):15-19.

-Programa de doctorado: Programa Oficial Universidad Autónoma de Madrid. Título de la tesis: Utilidad clínica del análisis de las células y las citoquinas en el lavado broncoalveolar de pacientes con disfunción crónica del injerto pulmonar. Fecha de titulación: 20/09/2017

We added the review of three recent publications about the implication of antifibrotic drugs for lung transplantation:

*The efficacy and safety of antifibrotic drugs in patients with advanced disease waiting for lung transplant is poor and is based on the analysis of clinical cases (22, 23, 24). Due to their antifibrotic action, both drugs could affect the surgical wound healing or cause anastomotic complications. In addition, nintedanib, as an inhibitor of tyrosine kinases platelet-derived growth factor and vascular endothelial growth factor receptors, may increase the risk of perioperative bleeding. Nevertheless, antifibrotic drugs doesn´t seem to increase post-transplant complications, such as dehiscence of the bronchial anastomosis, perioperative bleeding or graft dysfunction (25, 26). On the other side, the discontinuation of antifibrotic treatment in patients on the waiting list can worsen the clinical situation of the recipient. When slowing the progression of the disease with antifibrotic treatment, the patient is more likely to survive on the waiting list and get to transplantation in better condition*

-Lambers CBoehm PM1Lee SIus F3Jaksch PKlepetko WTudorache IRistl RWelte T, Gottlieb J.Effect of antifibrotics on short-term outcome after bilateral lung transplantation: a multicentre analysis. Eur Respir J. 2018 Jun 21;51(6).

- Balestro E, Solidoro P, Parigi P, Boffini M, Lucianetti A, Rea F Safety of nintedanib before lung transplant: an Italian case series.Respirol Case Rep. 2018 Mar 13;6(4):e00312

-Mortensen ACherrier LWalia R. Effect of pirfenidone on wound healing in lung transplant patients. Multidiscip Respir Med. 2018 Jun 14;13:16.

Round 2

Reviewer 1 Report

There continues to be many grammatical errors. I have placed comments on some of these. There are several; sweeping statements without referencing - some of these have been highlighted.

Author Response

 Thank you very much for your revision and suggestions. I attach manuscript with the last changes written in blue. 

*Due to their antifibrotic action, both drugs could affect the surgical wound healing or cause anastomotic complications* 

This is a hipotetical concern

Reviewer 3 Report

The Authors improved their manuscript and properly answered to my concerns, particularly inserting a paragraph regarding anti-fibrotic treatment in IPF patients

Author Response

 Thank you very much for your revision and suggestions